A non-linear optimization based robust attribute weighting model for the two-class classification problems

Alhudhaif Adi a.alhudhaif@psau.edu.sa
Department of Computer Science, College of Computer Engineering and Sciences in Al-kharj, Prince Sattam bin Abdulaziz University , Al-kharj , Saudi Arabia
Cunkas Mehmet
Electronic publication date: 2023 Sep 25
Publication date: 2023
Volume: 9
Electronic Location ID: e1598
Received 2023 Jul 27; Accepted 2023 Aug 28
Copyright: ©2023 Alhudhaif
Copyright year: 2023
Copyright holder: Alhudhaif
License: This is an open access article distributed under the terms of the Creative Commons Attribution License, which permits unrestricted use, distribution, reproduction and adaptation in any medium and for any purpose provided that it is properly attributed. For attribution, the original author(s), title, publication source (PeerJ Computer Science) and either DOI or URL of the article must be cited.
License URL: https://creativecommons.org/licenses/by/4.0/

Keywords: Nonlinear Attribute Weighting, Optimization, Machine Learning, Classification Problems

Funding: Deputyship for Research & Innovation, Ministry of Education in Saudi Arabia IF-PSAU-2021/01/18563 This work was supported by the Deputyship for Research & Innovation, Ministry of Education in Saudi Arabia through project number (IF-PSAU-2021/01/18563). The funders had no role in study design, data collection and analysis, decision to publish, or preparation of the manuscript.

==============================
Background

This article aims to determine the coefficients that will reduce the in-class distance and increase the distance between the classes, collecting the data around the cluster centers with meta-heuristic optimization algorithms, thus increasing the classification performance.

Methods

The proposed mathematical model is based on simple mathematical calculations, and this model is the fitness function of optimization algorithms. Compared to the methods in the literature, optimizing algorithms to obtain fast results is more accessible. Determining the weights by optimization provides more sensitive results than the dataset structure. In the study, the proposed model was used as the fitness function of the metaheuristic optimization algorithms to determine the weighting coefficients. In this context, four different structures were used to test the independence of the results obtained from the algorithm: the particle swarm algorithm (PSO), the bat algorithm (BAT), the gravitational search algorithm (GSA), and the flower pollination algorithm (FPA).

Results

As a result of these processes, a control group from unweighted attributes and four experimental groups from weighted attributes were obtained for each dataset. The classification performance of all datasets to which the weights obtained by the proposed method were applied increased. 100% accuracy rates were obtained in the Iris and Liver Disorders datasets used in the study. From synthetic datasets, from 66.9% (SVM classifier) to 96.4% (GSA Weighting + SVM) in the Full Chain dataset, from 64.6% (LDA classifier) to 80.2% in the Two Spiral datasets (weighted by BA + LDA). As a result of the study, it was seen that the proposed method successfully fulfills the task of moving the attributes to a linear plane in the datasets, especially in classifiers such as SVM and LDA, which have difficulties in non-linear problems, an accuracy rate of 100% was achieved.

Introduction

Machine learning solves various problems using data (Alpaydin, 2014). In machine learning, the measurable values of the observed event are called attributes (Chai et al., 2016; Bishop, 2006; Hedjazi et al., 2015; Yilmaz Eroglu & Kilic, 2017). Selecting the attributes independently from each other is an essential step in solving the problems. The independence of the attributes contributes positively to the classifier’s performance. However, it cannot be ensured that the attributes are always independent (Wettschereck, Aha & Mohri, 1997). Therefore, the class discrimination power of the classifier decreases (García, Luengo & Herrera, 2016). The different methods found in the literature aim to reduce the similarity of the attributes obtained from the observations with each other, in other words, to make the attributes independent.

Attribute weighting is a current issue in artificial intelligence studies. Although linear separators such as decision support machines (SVM) (Kim et al., 2003) or linear discriminant analysis (LDA) (Ye, Shi & Shi, 2009) have fast model generation properties in machine learning problems, they show low performance in problems where it is difficult to separate the attributes linearly. At this stage, the independence of the attributes that cannot be completely separated from each other arises. In the literature, attribute weighting methods are suggested to solve this problem. However, each method has its advantages and disadvantages. In this context, new weighting methods are needed to increase classification performance. Attribute (attribute) weighting is a data preprocessing method based on the idea that several attributes are more important (distinctive) than others to solve a classification problem. It should be expected that the contribution of each attribute is different in pattern classification. An attribute weighting algorithm allocates higher weight to related attributes while allocating less weight coefficient to less relevant and unnecessary attributes. A general classification model and the addition of the weighting process to this model are shown in Fig. 1 (Niño Adan et al., 2021). It can be expressed by a general classification process, data acquisition, extraction of attributes, and classification process. The attribute weighting process is added before the classification step. With weighting, the property of the attribute is not changed. The only revaluation is done to allow the data to be viewed differently.

Figure 1 Attribute weighing block adding to the general machine learning system.

An example drawing showing the different views obtained by weighting the attributes is given in Fig. 2. The figure shows two attribute transformations for two classes. Data points belonging to the 1st class are indicated with an asterisk (*), and data points belonging to the second class are marked with the “+” sign.

Figure 2 Transform of attributes after attribute weighting method.

Data points belonging to the 1st class are indicated with an asterisk (*), and data points belonging to the second class are marked with the “+” sign.

As seen in the literature review, and different mathematical methods, rule-based methods, and heuristic methods are used to select weighting coefficients. However, each has different advantages and disadvantages.

Many studies have been done on attribute weighting problems, and some of them are detailed below designed an automated decision support system for breast cancer diagnosis. As the classifier, they used a neural network, and as data preprocessing, they proposed a new attribute weighting method based on Ant Lion optimization. Also, their work used the Tanh method for data normalization (Dalwinder, Birmohan & Manpreet, 2020). Alimi et al. (2021) proposed a new hybrid model to classify unwanted power events. This proposed hybrid model consists of two stages. In the first stage, they proposed an attribute weighting method based on the evolutionary genetic algorithm (GA) model, and the attributes were optimally weighted according to the wrapper method. Next, they classified the unwanted power events with the support vector machine (SVM) (Alimi et al., 2021). Dialameh & Jahromi (2017) proposed a dynamic attribute weighting method based on order and distance between data. They used the Nearest neighbor (NN) as classifier and obtained promising results by combining it with the preprocessing method they developed. Daldal, Polat & Guo (2019) proposed a new attribute weighting method based on Neutrosophic c-means (NCM) to classify digital modulation signals. First, they used Linear Discriminant Analysis (LDA), Support Vector Machine (SVM), k-nearest neighbor (k-NN), AdaBoostM1, and Random Forest to classify the raw data. Later, they obtained excellent results in determining digital modulation types by weighing the attributes obtained from the signals. Kim (2018) proposed a new semi-consulting size reduction algorithm based on the attribute weighting method in sentiment analysis. He used six different datasets in his study and obtained different results for each dataset. They have obtained high-performance results thanks to the attribute weighting method (Kim, 2018). Kim proposed a hybrid method to identify fault detection in rolling bearings. In this hybrid method, the first extracted attributes in the time and frequency domain from the vibration signals. Then, they performed the attribute selection process with correlation analysis and principal component analysis-weighted load evaluation method. They also used sensitivity analysis to weight attributes. Finally, they used a particle swarm optimization-support vector machine (PSO-SVM) to classify the error conditions and obtained good results (Li, Dai & Zhang, 2020). Ruan et al. (2020) proposed a new attribute weighting method for Naïve Bayes text classifiers. In their weighting approach, a weighting coefficient was obtained for each separate class and applied to the text classification problem. They combined the class-specific deep learning-based attribute weighting method with the Multinomial Naïve Bayes classifier and applied it to the text classification problem (Ruan et al., 2020). Jiang et al. (2019) proposed a new data preprocessing algorithm to improve the performance of the naive Bayes classifier. They proposed a new correlation-based attribute weighting method by improving the minimum redundancy maximum relevance (MRMR) algorithm (Jiang et al., 2019). Panday, de Amorim & Lane (2018) proposed a new attribute weighting method for attribute selection without consulting. They developed a set-dependent attribute weighting method that ed the attributes’ interest and applied it to different datasets (Panday, de Amorim & Lane, 2018). Tao, Cao & Liu (2018) proposed a new dynamic attribute weighting method for recommender systems based on user preference sensitivity. They have achieved over 81% success with their proposed system (Tao, Cao & Liu, 2018). The study uses the metaheuristic method to calculate the weights that will make the attributes independent from each other with a mathematical model.

The second chapter of the study explains the datasets used, the proposed mathematical model, and the metaheuristic algorithms used. In the third chapter, the distributions of the data are rearranged with the coefficients obtained because of running the algorithms, and the results of the classification algorithms are given. Finally, the article ends with a discussion in the fourth section and conclusions in the fifth section.

Materials and Methods

A new mathematical model is proposed to find the weighting coefficients in the study. Optimization algorithms optimization algorithms provide the solutions of the proposed model provide the solutions of the proposed model. The proposed model is a novel attribute weighting method. Our mathematical model, detailed in the methodology section, analyzes the differences in data distribution in general terms, processes the attributes belonging to the same class, and brings the cluster centers closer. At the same time, the attributes that point to different classes are separated. In this respect, attributes are processed at both in-class and inter-class levels. This process is carried out by examining the change of the attributes belonging to the same class within itself and the change of these attributes according to other classes: the deviations in the data distribution. The problem at this stage is determining the coefficients that will make the changes in the data distribution. Optimization algorithms have been used as a solution. To reveal the consistency of the results obtained from the study, it was tested with real and artificial data with two or more attributes and classes. In addition, the proposed method has been tested with four different algorithms: the particle swarm algorithm, the bat algorithm, the flower pollination algorithm, and gravitational search algorithm. A block diagram of the proposed method is given in Fig. 3. In this section, the firstly used datasets will be explained. Then, after briefly mentioning the optimization algorithms, the proposed mathematical model will be explained. The classifiers used in general terms will be mentioned in the last section.

Figure 3 The block diagram of the proposed method.

Dataset

Four separate datasets were used in the study. Two are the Liver Disorders dataset and the Iris dataset obtained from real observations, while the Full Chain dataset and Two Spiral dataset are artificial datasets. Linear classifiers and sets that are difficult to distinguish were preferred in selecting artificial datasets. All datasets consist of real numerical attributes.

Liver Disorders dataset

The Liver Disorders dataset was prepared by Bupa Medical Research. It is a dataset consisting of six attributes, five of which are attributes obtained by blood tests and one of which is the amount of alcohol consumption. It contains a total of 345 observations. It has positive and negative classes (BUPA Medical Research Ltd, 1990; McDermott & Forsyth, 2016). In Fig. 4, given data distribution for three attributes. The figure shows that the class represented by red crosses and blue circles are intertwined.

Figure 4 Data distribution of the Liver Disorder dataset for three attributes.

Iris dataset

The Iris dataset is frequently used in machine learning studies. The dataset includes 150 observations of three different iris plants. It has four attributes. The species names with class output are numbered 0 − 1 − 2 (Fisher, 1936; Fisher, 1988).

Full chain dataset

The Full Chain dataset is an artificially generated dataset with two nested rings. The dataset we produced for the study contains three attributes and two classes. One thousand observations were produced. In Fig. 5, data distribution for three attributes is provided. The figure shows that the class represented by red crosses and blue circles is in a nested ring structure.

Figure 5 Data distribution of the Full Chain dataset for three attributes.

Two Spiral dataset

The Two Spiral dataset is an artificial dataset produced as two spirals nested within each other. It was produced as 2,000 observations with two attributes and two classes. In Fig. 6, data distribution for two attributes is provided. The figure shows that the class represented by red crosses and blue circles is in a spiral structure.

Figure 6 Data distribution of Two Spiral dataset for two attributes.

Optimization algorithms

This section briefly explains the optimization algorithms used in the study. Each algorithm looks for possible solutions with the fitness function described in the mathematical model section. Solutions close to the best possible result are marked as the best. In the study, optimization algorithms look for weight values. Therefore, the initial dimensions are equal to the product of the total number of attributes (m) and the number of classes (C) as given in Eq. (1). Each algorithm uses the initial values given in Table 1. The literature’s most used common values are used for initial values.

Table 1 Initial parameters for all optimization algorithms.

	Algorithm	PSO	FPA	BAT	GSA	
Initial Parameters	Iteration	100	
Solution	20	
Upper Limit	1	
Lower Limit	−1	
c1	2.1				
c2	1.9				
w	0.8				
p		0.8			
A			1		
r0			1		
alpha			0.97		
gamma			0.1		
ɛ				2−52	
Notes.

PSO Particle swarm optimization algorithm

FPA flower pollination algorithm

BAT bat algorithm

GSA gravitational search algorithm

(1) Dimension=m∗C

As can be seen in Table 1, the number of iterations, which are the stopping criteria, the number of solutions to be calculated in a single iteration, and the lower and upper limits for the weighting coefficients are given jointly for the four algorithms. The specific starting parameters of each algorithm are explained in the headings below.

Particle swarm optimization algorithm (PSO)

The particle swarm algorithm is an algorithm that models the movements of animals living in a herd during foraging (Arican & Polat, 2020). In the PSO algorithm, each particle and the structure created by them is called a flock. Among the initial parameters given in Table 1, c1 is the particle coefficient, and c2 is the social coefficient. It determines the effect of individual and herd on total movement. w is the moment of inertia, which causes each particle to move somewhat, even if it has the best result.

Bat algorithm (BAT)

The bat algorithm is a metaheuristic algorithm that models the way-finding behavior of bats with resonance (Yang, 2010a). From the initial conditions given in Table 1A shows the sound intensity, and r0 shows the propagation rate of the signal. The values of a and are constant values. Selection is made according to the (0) <a <1 and γ>0.

Gravitational search algorithm (GSA)

It is a heuristic algorithm modeled on the basis of Newton’s laws of gravity and motion (Rashedi, Nezamabadi-pour & Saryazdi, 2009). Each piece in the search space is considered a mass, so it is an artificial mass system. All masses exert a force of attraction to each other in the search space and move in the search space under the influence of these forces. The ɛ given in Table 1 refers to a small, fixed number entered by the user.

Flower pollination algorithm (FPA)

FPA is a metaheuristic model based on flowering plants’ pollination and reproduction process (Yang, 2014). It uses biotic reproduction, the reproductive characteristic of flowering plants globally, and abiotic reproduction, as local exploration. P given in Table 1 is a fixed probability value between 0 and 1 for local and global pollination.

The mathematical model of the optimization-based dynamic attribute weighting model

The mathematical model used in determining the weighting coefficients in the study will be explained in this section. At the same time, this model is the fitness function of optimization algorithms.

An example of in-class and between-class distance measurements is given in Fig. 7 for two classes and two attributes. Here dc shows the Euclidean distances of the attribute pairs belonging to the tenth observation for the o c’th grade to their class center. d’c, o shows the Euclidean distances of the attribute pairs belonging to the observation number o for class c to other class centers. Figure 8 gives the flowchart of the proposed model.

Figure 7 (A–B) Demonstration within and between class distances in the proposed model.

Figure 8 The flowchart of the proposed model.

Equation (2) calculates the sum of the averages of the distances for the whole class. Here O gives the number of in-class observations. C gives the total number of classes. (2) WithinClassDistance= ∑c=1C ∑o=1Odc,oOc

With Eq. (3), the sum of the averages of the distances between all classes is calculated. (3) BetweenClassDistance= ∑c=1C ∑o=1Odc,o′Oc

Optimization algorithms looking for weight values by using the fitness function in Eq. (4) perform the weighting process with the equation given in Eq. (5). Here m is the total number of attribute s, and C is the total number of classes. (4) fitness=WithinClassDistanceBetweenClassDistance

Optimization algorithms looking for weight values using the fitness function in Eq. (4) perform the weighting process with the equation given in Eq. (5). Here m is the total number of attribute s, and C is the total number of classes. w1∗f1,w2∗f2,…,wm∗fm

wm+1∗f1,wm+2∗f2,…,w2m∗fm

(5) ⋮

vwn−m+1∗f1,wn−m+2∗f2,…,wn∗fm

n=C∗m

Classifiers

Support vector machine and linear discriminant analysis, which perform linear classification, were preferred to see the effect of weighting attributes independently of the classifier. Thus, it can be shown when problems are solved linearly. This section briefly mentions the classifiers; detailed information can be viewed from the references.

Support vector machine (SVM)

It is a supervised learning system used in classification problems. It uses a linear line found at the maximum distance to the data of the two classes to separate the classes. Due to its linearity, its performance decreases in nested datasets (Kim et al., 2003; Sharma et al., 2023; Nour, Senturk & Polat, 2023). SVM are a supervised learning method used in classification problems. Draws a line to separate points placed on a plane. It aims to have this line at the maximum distance for the points of both classes. It is suitable for complex but small to medium datasets. To make the classification, a line separating the two classes is drawn, and the region between ±1 of this line is called the margin. The wider the margin, the better the separation of two or more classes (Yang, 2010b). Figure 9 shows the flowchart of the SVM model.

Figure 9 The flowchart of the SVM model.

For a dataset with n elements; (6) x1,y1,………,xn,n,x∈Rd,x∈+1,−1

The training data can be separated with the coefficients w and w0 and the separator plane to be provided by the equation given in Eq. (7). (7) Dx=w×x+w0

The separating plane must fulfill the conditions given in Eq. (8). (8) Dxi=w×x+w0≥+1,ifyi=+1w×x+w0≤−1,ifyi=−1i=1,2,…,n

Equation (9) is obtained by combining Eq. (8). (9) yiw×xi+w0≥+1,i=1,2,……,n

The distance between the dividing plane and the nearest vertex is called the boundary and is denoted by Γ. The larger the limit value, the better the classifier generalizes. The condition given by Eq. (10) must be satisfied for all training examples. (10) yk×Dxkw≥Γ,k=1,2,……,nveyk∈−1,+1

The condition that must be met for the best discrimination value is to set the limit value Γ to be the highest. Because the weight w can take an infinite number of values. The data points on the boundary are called support vectors.

Linear discriminant analysis (LDA)

LDA looks for vectors that best separate classes rather than the one that separates data best. It creates a linear combination that gives the largest mean differences between the desired classes. It is defined by a score function defined by Fisher (Ye, Shi & Shi, 2009).

It was first developed by Fisher (Sayad, 2019). The analysis defines the score function given in Fisher Eq. (12). The problem in this equation is to find the coefficients that will maximize the score. (11) Z=β1x1+β2x2+.…….+βnxn

(12) Sβ=βTμ1+βTμ2βTCβ

(13) Sβ=Z1¯−Z2¯GruplardakiZvaryansı

(14) β=C−1μ1−μ2

(15) C=1n1−n2n1C1−n2C2

Here β shows the model coefficients, C covariance matrices, C mean vectors.

Figure 10 (A–D) Fitness graphs at the end of each iteration for the Iris dataset.

Results

The proposed method has been tested on four datasets, two of which are artificial. To see the contribution of the proposed weighting method to the classification performance, SVM and LDA classifiers that classify in a linear plane were preferred.

In the first part, the distribution of the unprocessed attributes of each dataset and the distributions of the weighted attributes will be given. In the second part, the classification results of the raw attributes and the attributes obtained as a result of the weighting process will be shared. In the last part, the results will be compared. Measurements of the complex matrix have demonstrated classification performance.

Data distributions

Iris dataset

In Fig. 10, the graph of the fit functions calculated by each optimization algorithm in iterations is given. Figures 10A–10D, respectively, refer to the PSO, BAT, FPA, and GSA algorithms.

Liver Disorders dataset

Figure 11 shows the data distributions of the Liver Disorders dataset. In Fig. 11, blue and red-colored icons show data for each class. Figure 11A shows the data distribution multiplied by the weights obtained by the PSO algorithm. Similarly, the data distributions were obtained by the BAT algorithm (Fig. 11B), the FPA algorithm in (Fig. 11C), and the GSA algorithm in (Fig. 11D). When the distribution of the weighted data is examined, it is seen that the data approach each other within the class and diverge between the classes. Figure 12 shows the graph of the fit functions calculated by each optimization algorithm in iterations. Figures 12A–12D, respectively, refer to the PSO, BAT, FPA, and GSA algorithms.

Figure 11 Data distribution for the Liver Disorders dataset.

Figure 12 Fitness graphs at the end of each iteration for the Liver Disorders dataset.

Full Chain dataset

Data distributions of the Full Chain dataset are shown in Fig. 13. In Fig. 13, blue and red-colored icons show data for each class. Figure 13A shows the distribution of the data multiplied by the weights obtained by the PSO algorithm. Similarly, the data distributions were obtained by the BAT algorithm in (Fig. 13B), the FPA algorithm in (Fig. 13C), and the GSA algorithm in (Fig. 13D). When the distribution of the weighted data is examined, it is seen that the data approach each other within the class and diverge between the classes. Figure 14 gives the graph of the fit functions calculated by each optimization algorithm in iterations. Figures 14A–14D, respectively, refer to the PSO, BAT, FPA, and GSA algorithms.

Figure 13 Data distribution for Full Chain dataset.

Figure 14 Fitness graphs at the end of each iteration for the Full Chain dataset.

Two Spiral dataset

Data distributions of Two Spiral datasets are shown in Fig. 15. In Fig. 15, blue and red-colored icons show data for each class. Figure 15A shows the distribution of the data multiplied by the weights obtained by the PSO algorithm. Similarly, the data distributions were obtained by the BAT algorithm (Fig. 15B), the FPA algorithm (Fig. 15C), and the GSA algorithm (Fig. 15D). When the distribution of the weighted data is examined, it is seen that the data approach each other within the class and diverge between the classes. Figure 16 gives the graph of the fit functions calculated by each optimization algorithm in iterations. Figures 16A–16D, respectively, refer to the PSO, BAT, FPA, and GSA algorithms.

Table 2 Classification accuracy rate for PSO algorithm (%).

Dataset	Non-Weighted	Our weighted algorithm	
	LDA	SVM	Optimization	LDA	SVM	
Iris	98.0	97.3	PSO FPA BAT GSA	100.0 100.0 100.0 100.0	100.0 100.0 100.0 100.0	
BUPA Liver Disorders	69.0	68.4	PSO FPA BAT GSA	100.0 100.0 100.0 100.0	100.0 100.0 100.0 100.0	
Full Chain	66.6	66.9	PSO FPA BAT GSA	82.8 82.8 70.5 82.7	95.2 92.2 70.2 96.4	
Two Spiral	64.6	65.3	PSO FPA BAT GSA	79.8 79.3 80.2 79.6	69.2 69.2 71.4 69.5	

Figure 15 (A–D) Data distribution for Two Spiral dataset.

Figure 16 (A–D) Fitness graphs at the end of each iteration for Two Spiral dataset.

Experimental results

Table 2 gives the accuracy rates for all datasets and weighting processes. As can be seen, when the results obtained were examined, an increase was observed in the performances of all datasets. Especially in the real dataset Iris and the Liver Disorders dataset, a 100% accuracy rate has been reached. Furthermore, accuracy rates exceeding 95% from 65% to −70% have also been achieved in artificial datasets. Therefore, regardless of the algorithm used, it is considered appropriate to determine the weighting values of the proposed mathematical model.

Discussion

As a result of the study, it was seen that the proposed method successfully fulfills the task of moving the attributes to a linear plane in the datasets. Especially in classifiers such as SVM and LDA, which have difficulties in non-linear problems, an accuracy rate of 100% was achieved. Since the proposed weighting method determines the weights with heuristic and meta-heuristic-based optimization algorithms, each dataset is handled in its way. This situation contributes positively to the increase in performance. The so-called real and artificial datasets with different attributes have been seen in other applications. The method suggested for multi-class and multi-attribute problems has increased performance in all datasets. In light of all these findings, the proposed method can be used in classification problems with the weighting method.

Conclusions

Since the proposed model is mathematically based, it only allows operations on accurate numerical data. Therefore, it cannot be used directly in, categorical data text, or categorical datasets such as YES/NO. Datasets containing such attributes should first be converted to numerical data if possible. Otherwise, since calculations such as cluster center and distance cannot be performed, obtaining a coefficient or processing the attributes with a coefficient is impossible. The Euclidean distance used in the study is a fundamental metric, and its effects on success in other metrics should be observed. As the number of attributes increases, the number of weights to be calculated increases, so the time to calculate the fit value of the optimization algorithm will increase.

Supplemental Information

Supplemental Information 1 Matlab code

Click here for additional data file.

Additional Information and Declarations

Competing Interests

Author Contributions

Data Availability

The author declares that they have no competing interests.

Adi Alhudhaif conceived and designed the experiments, performed the experiments, analyzed the data, performed the computation work, prepared figures and/or tables, authored or reviewed drafts of the article, and approved the final draft.

The following information was supplied regarding data availability:

The data are available at the UCI Machine Learning Repository:

- Liver Disorders. (1990). UCI Machine Learning Repository. https://doi.org/10.24432/C54G67.

- Fisher, R. A.. (1988). Iris. UCI Machine Learning Repository. https://doi.org/10.24432/C56C76.

The Matlab code is available in the Supplemental Files and at Github: https://github.com/kpolat14/data-weighting-model-and-classification.

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
