# Peer review of "A non-linear optimization based robust attribute weighting model for the two-class classification problems"

_PeerJ Computer Science, doi:10.7717/peerj-cs.1598_

## Round 0.1 · original submission · Major Revisions

The author should make the necessary revisions to the article in order to improve its overall quality, clarity, and impact. Specifically, they should focus on:

- Strengthening the introduction by emphasizing the contributions and motivation behind the work.

- Expanding on the training and testing phases of each dataset to enhance reproducibility.

- Introducing a dedicated "Limitations and Future Work" section in the conclusion to address study limitations and suggest potential future research directions.

- Incorporating recent references.

- Clearly highlighting the novelty and contributions of the proposed method, particularly in the introduction.

- Clarifying the motivation behind the paper to engage readers effectively.

- Explicitly addressing the study's limitations to provide a comprehensive context for interpreting the results.

**Language Note:** The review process has identified that the English language must be improved. PeerJ can provide language editing services - please contact us at copyediting@peerj.com for pricing (be sure to provide your manuscript number and title). Alternatively, you should make your own arrangements to improve the language quality and provide details in your response letter. – PeerJ Staff

Reviewer 1 ·

Basic reporting

• The article should be checked for grammar.
• The contributions of the work given in the introduction to this paper can be improved.
• Some tables and figures in the “Experimental Analysis and Discussions” section need more explanations and discussions.
• Add more details about each dataset's training and testing phases.
• Limitations and future work are missing in the last section: “Conclusion”. Please add these two parts to this section.
• It will be much better to include some references published from 2021 to the present.

Experimental design

The design is good. There are some good explanations showing the parameters.

Validity of the findings

Tables and figures are given in appropriate logical order.

Additional comments

• The article should be checked for grammar.
• The contributions of the work given in the introduction to this paper can be improved.
• Some tables and figures in the “Experimental Analysis and Discussions” section need more explanations and discussions.
• Add more details about each dataset's training and testing phases.
• Limitations and future work are missing in the last section: “Conclusion”. Please add these two parts to this section.
• It will be much better to include some references published from 2021 to the present.

Reviewer 2 ·

Basic reporting

The author suggested a hybrid approach to classify the medical and artificial datasets.
In the paper:
“In the study, the location of the cluster centers belonging to the classes was taken into account. While it is ensured that all in-class observations are gathered around the cluster center, it is also aimed to move the cluster centers of different classes away from each other. A mathematical model is proposed that can determine these weights.”

Experimental design

There are two classes datasets in the simulations.

Validity of the findings

No comment

Additional comments

a) The English of this paper should be polished.
b) Some points in Figure 1 should be corrected.
c) Did authors use real-time data?
d) Please add more discussion.
e) Please give an example showing the proposed system work.
f) the caption of figure10 should be checked

Reviewer 3 ·

Basic reporting

The manuscript has been organized well. The language of the manuscript is clear. The required literature summary and material and methods have been presented appropriately. I found the paper to be somewhat interesting. However, the below issues should be addressed if the authors would like to pursuit its publication.
1. The name of the used open-access database should be specified in the abstract section.
2. First, the paper's contribution to expert and intelligent systems should be clearly outlined. The novelty of the proposed method should be highlighted. The authors should clarify the paper's contributions in the introduction section.
3. In the introduction, the motivation of the paper needs to be articulated far more clearly.
4. Furthermore, where are the limitations of your study? Clarifying the limitations of a study allows the readers to understand better under which conditions the results should be interpreted.

Experimental design

To validate the proposed models, the authors have made some experiments. The structures are good.

Validity of the findings

See above.

Additional comments

See above.

---

## Round 0.2 · accepted · Accept

Necessary revisions have been made in line with the referee's suggestions.

Reviewer 1 ·

Basic reporting

it is done

Experimental design

It is done

Validity of the findings

It is done

Additional comments

Accepted

Reviewer 2 ·

Basic reporting

Its done, the revised version of the paper looks good

Experimental design

Its done, the revised version of the paper looks good

Validity of the findings

Its done, the revised version of the paper looks good

Reviewer 3 ·

Basic reporting

I appreciate the author's effort to improve the article. The article is acceptable as it is.

Experimental design

I appreciate the author's effort to improve the article. The article is acceptable as it is.

Validity of the findings

I appreciate the author's effort to improve the article. The article is acceptable as it is.

Additional comments

I appreciate the author's effort to improve the article. The article is acceptable as it is.